# Preference-Tree-Based Real-Time Recommendation System

**DOI:** 10.3390/e24040503

**Published:** 2022-04-02

**Authors:** Seongju Kang, Kwangsue Chung

**Affiliations:** Department of Electronics and Communications Engineering, Kwangwoon University, Seoul 01897, Korea; sjkang@cclab.kw.ac.kr

**Keywords:** recommendation systems, information filtering, data sparsity, cold start, preference tree, real-time requirements

## Abstract

In the current era of online information overload, recommendation systems are very useful for helping users locate content that may be of interest to them. A personalized recommendation system presents content based on information such as a user’s browsing history and the videos watched. However, information filtering-based recommendation systems are vulnerable to data sparsity and cold-start problems. Additionally, existing recommendation systems suffer from the large overhead incurred in learning regression models used for preference prediction or in selecting groups of similar users. In this study, we propose a preference-tree-based real-time recommendation system that uses various tree models to predict user preferences with a fast runtime. The proposed system predicts preferences based on two balance constants and one similarity threshold to recommend content with a high accuracy while balancing generalized and personalized preferences. The results of comparative experiments and ablation studies confirm that the proposed system can accurately recommend content to users. Specifically, we confirmed that the accuracy and novelty of the recommended content were, respectively, improved by 12.1% and 27.2% compared to existing systems. Furthermore, we verified that the proposed system satisfies real-time requirements and mitigates both cold-start and overfitting problems.

## 1. Introduction

Current media content platforms provide large amounts of content to users, and users can utilize keywords to search for preferred content or consume the content presented by service providers. However, users typically use only narrow content categories with specific keywords to avoid excessive content appearing in the search results. Thus, information overload severely affects the utilization efficiency of online data [1]. Recommendation systems seek to address this issue by selecting and presenting only content that may be of interest to the user. A personalized recommendation system provides content based on various types of information, such as the browsing history and the video watch history of the user.

Early recommendation systems expressed the relationship between users and content as a matrix and predicted similarity, relevance, and preferences [2,3,4,5]. The user-based filtering recommendation system recommends contents to a target user by predicting the preferences of other users with a high similarity to those of the target user. Content-based filtering (CBF) predicts the similarities between contents and recommends contents with high similarities to the user’s preferred contents. This filtering-based method predicts similarity with a high accuracy using simple operations. However, it functions by performing matrix operations with large time and space complexities and consequently has scalability problems. Furthermore, it experiences data sparsity problems for new users and contents for which there is insufficient related information [6,7,8].

The collaborative filtering (CF) approach avoids the problems faced by the matrix-based approach by considering the preferences of both the target user and multiple users. In the CF approach, contents are recommended to a target user based on the historical data of similar users [9]. The CF-based recommendation system clusters existing and new users based on user profiles, such as gender, region, and age. Furthermore, it updates similar groups based on the user’s history to mitigate the data sparsity problem. However, as the initial preference group for the new user is determined only by the user’s interactions, the cold-start problem may occur if the initial recommendation is incorrect.

Several researchers have focused on solving the cold-start problem in recommendation systems. However, existing preference prediction schemes are limited in their consideration of various factors. With the development of deep-learning (DL) algorithms, the deep neural network (DNN) model was introduced into recommendation systems. For example, Chi et al. [10] trained a regression model with a recommendation system based on matrix factorization (MF) to predict unknown preferences between users and content. MF-based systems can consider various characteristics by learning from the histories of all users. Additionally, owing to the characteristics of the regression model, recommendations can be made with a fast runtime. However, because the MF method is time-invariant, the DNN model has to be retrained for it to consider new historical data. Furthermore, the data sparsity problem occurs when the amount of data for model training is insufficient, which degrades the accuracy of the model.

To predict preferences considering real-time user interactions, a time-variant recommendation system is required. Such a recommendation system must be updated in real time to predict user preferences, and a non-fixed scalable model is required to incorporate various factors. In this study, we propose a recommendation system that can predict real-time preferences based on a preference tree. Specifically, we introduce a tree model that predicts user preferences and enables the recommendation system to update user preferences in real time, thereby achieving a time-variant recommendation system. This study’s main contributions are as follows:A time-variant recommendation system is proposed. The proposed system generates preference trees based on a user’s history, which are then used to create a collaborative similar graph (CSG). The CSG is then used to define similarity groups to solve the data sparsity problem for users with insufficient historical data.Generalized preferences are considered by creating a federated tree for the entire user history to solve the cold-start and data-sparsity problems.

The remainder of this paper is organized as follows. Section 2 presents related work and discusses the limitations of existing recommendation systems. Section 3 describes the proposed real-time recommendation system. Section 4 presents the performance evaluation of the proposed recommendation system. Finally, Section 5 presents concluding remarks.

## 2. Related Work

### 2.1. Recommendation Systems

Recommendation systems help users select appropriate content from a wide range of applications [11,12]. Several algorithms have been proposed on the basis of information filtering schemes that recommend contents to users by considering relevant information. As illustrated in Figure 1, recommendation systems can be classified into two main approaches: equation-based and DL-based systems. Equation-based systems formulate a similarity prediction problem based on a user’s history and implicit information. Examples include the CBF, CF, and hybrid-filtering (HF) methods. DL-based systems leverage a DNN model and train a regression model using supervised learning to predict the similarities and relationships between users and contents, considering all user history data.

CBF has been leveraged to recommend contents on the basis of historical data, such as rating scores, numbers of uses, and review data [13,14,15]. In CBF, content preferences are predicted by inferring relationships among users or between users and contents. To infer relationships among contents, systems should be aware of the contents’ categorical properties. Term frequency–inverse document frequency (TF-IDF) methods extract keywords that estimate categorical weight based on the frequency of appearance [16]. As CBF uses historical data for recommendations, sparsity and overfitting problems are commonplace. If the amount of historical user data is sparse, the recommendation accuracy may degrade. However, CBF generally recommends only contents that are similar to those purchased or liked by the user in the past, as long as they are concentrated into a specific category [17].

To overcome the shortcomings of CBF, several additional factors should be considered. To resolve overfitting, both user history and the preferences of similar user groups should be considered [18,19,20]. To identify similar groups, CF uses various techniques, such as Pearson’s correlation coefficient (PCC), cosine similarity, and Euclidean distance. Koohi et al. [21] proposed a user-based CF fuzzy C-means clustering method for this purpose and calculated the similarity using cosine similarity. Further, Tan et al. [22] proposed resonance similarity based on three distance factors. Genetic algorithms [23] and other innovative equations [24], such as mean-squared difference, have also been applied. HF recommendation systems combine CBF and CF schemes to overcome the drawbacks of both [25,26]: CBF schemes overcome the cold-start problem, while CF ensures recommendation accuracy.

Several DL-based recommendation systems have also been investigated [27,28]. MF, for example, learns the latent factors of users and contents by considering a historical user-content matrix to predict interactions. Low-dimensional (low-rank) MF models are popular as they generate the most accurate predictions [29]. Yi et al. [30] proposed a deep MF framework that creates a graph based on the user’s historical and implicit feedback data to calculate similarity via a maximum-likelihood estimation between the user and content. Chen et al. [31] designed a deep attention-based logistic-regression model for CF in which a low-dimensional rating matrix between users and contents is learned to address model limitations [32]. However, because DL-based schemes are time-invariant, they require additional DNN training to learn new historical data. Notably, insufficient training data may result in data-sparsity and cold-start problems.

### 2.2. Recommendation in Real-Time Systems

Existing recommendation systems manage user and content data in a matrix and apply matrix operations with large space and time complexities to make predictions. Such systems are unsuitable for real-time predictions because complexity increases remarkably with numbers of users and contents. Zhang et al. [33] mitigated quantization loss using a discrete-factorization machine-learning model. He et al. [34] proposed fast MF schemes to reduce training and prediction runtimes. Further, Li et al. [35] proposed an all-weighted fast optimization scheme in which a learning algorithm based on element-wise alternating least squares is applied to optimize the model using variably weighted missing data. Chen et al. [36] designed a greedy posterior maximum-inference algorithm to mitigate the NP-hard nondeterministic drawbacks of real-time prediction. However, no solutions yet exist that can enable matrix operations to support real-time prediction.

Several existing systems recommend top-N content based on high ratings. However, such methods are unsuitable for real-time prediction because they predict preferences based on historical and implicit data when users are offline. Therefore, a novel data model is required to replace the matrix construct. Recently, Zhu et al. [37] proposed a tree-based deep model (TDM) that predicts preferences using iterative learning and a maxheap-like tree probability formulation. However, it requires fixed-sized trees; hence, it often fails to classify content properties in detail. In response, Mu et al. [38] and Rathore and Sandeep [39] proposed decision-tree-based recommendation systems that recommend decisions based on predefined rules. However, in this case, comprehensive user characteristics cannot be considered.

In this study, we introduce a tree model to solve the high computational complexity and low-scalability problems of previous recommendation systems. Additionally, a novel preference-prediction algorithm is proposed to satisfy the real-time requirements of the time-variant system.

## 3. Proposed System

In this section, we introduce the overall framework of the proposed recommendation system and tree model used for real-time recommendations. Additionally, we describe the methods and equations used to predict user preferences and propose a recommendation algorithm based on the federated tree and CSG, which are introduced to solve the problems of the existing recommendation systems.

### 3.1. Proposed Recommendation System

Existing recommendation systems represent all elements attributed to users and their content in a matrix. In the case of CF, a new vector must be included for the corresponding information when a new user or new content is added. Therefore, existing CF systems have scalability limitations. Additionally, DL-based recommendation systems have limitations in personalized recommendations because they predict preferences based on generalized weights for all users. To address these drawbacks, we propose a lightweight and scalable recommendation system based on a tree model that considers both personalized and generalized preferences. Figure 2 illustrates the architecture of the proposed system. As indicated in the figure, a personalized tree is generated based on the historical data of each user, and a federated tree is created to predict the generalized preference. The proposed system creates a CSG to mitigate the cold-start problem that may occur for new users. Furthermore, it predicts user preferences for content based on three types of trees, specifically federated, personalized, and similarity trees.

### 3.2. Preference Tree Model

To enable real-time recommendations, we propose a preference-tree model that manages data adaptively, making it suitable for representing hierarchical, categorical content structures and retrieving attributes within a logarithmic time complexity, making it more suitable than a matrix. It further reduces computational costs by merging or discarding redundant or unused data. Figure 3 depicts the proposed structure. The root node contains profile data for identifying users, and all internal and leaf nodes comprise attribute-based content information reflecting the user’s historical data. Each category is divided into subcategories, and the longest common-category (LCC) node [40] is used to predict similarity. As categories are defined based on content properties, the higher the similarity between contents, the longer the LCC node.

To infer user preferences based on new content data, conventional systems require additional processing (e.g., feature-map analyses, similarity comparisons, and model training). In contrast, the proposed system predicts preferences for new content by determining the user’s LCC node in the preference tree. To achieve this with a low time complexity, the tree applies an ordered HashMap data structure, which stores key–value paired data wherein all pairs are sorted based on the key value. For example, *Node A* is stored as the pair *<Sub Category 2, Node A>* in the HashMap of the root node. When the data are sorted, the target data are retrieved using a binary-tree search, which implies a logarithmic time complexity. Matrix-based searches require an O(N) time complexity. Figure 4 depicts the time complexity of the LC with respect to the number of contents, N, resulting in a $O\left(\log_{2}N\right)$ complexity. To depict the worst case, assume that the LCC performs similarity comparisons at every node. Hence, a maximum $O\left(\log_{2}N\times \log_{2}N\right)$ operation is applied. If N>1000, a significant performance difference begins to appear when comparing matrix performances.

### 3.3. Tree Model Formulation

The proposed recommendation system generates a personalized preference tree using historical data. The tree model contains nodes containing the attributes of hierarchical, categorical contents. Table 1 describes the node components. We consider c to be a category name mapped into a short string to reduce the comparison complexity, and $n_{c}$ is the number of *c*s in the user’s historical data. $N_{child}$ indicates the number of child nodes of node c, and d denotes depth. The value of d begins from zero at the root node and is incremented by one as it moves toward the child node. Therefore, d reflects the level of detailed granularity and c represents content characteristics. $d_{max}$ denotes the largest $d_{c}$ of the descendant node and represents the maximum granularity of the category. We consider $P_{c}$ to be the preference for node c.

We calculated $P_{c}$ using Equation (1), where $\frac{d_{c}}{d_{max}}$ and $\frac{n_{c}}{n_{d}}$ represent the normalized terms for the depth and breadth, respectively. These two expressions were adopted to improve the accuracy of the recommendations. α denotes a constant value for the depth or breadth depending on whether the preference tree considers personalized characteristics or novelty. To consider the recent user preferences, historical data are divided into s sections and assigned different weights; $w_{i}$ denotes the weight of the ith section.
(1)$$P_{c}=\overset{N}{\underset{i=1}{\sum }}\left(\left(1-\alpha \right)\times \frac{d_{c}}{d_{max}}+\alpha \times \frac{n_{c}}{n_{d}}\right)\times w_{i}$$

### 3.4. Federated Tree

The personalized preference tree predicts the preference for content based on the user history. However, cold-start and data-sparsity problems may occur for a new user with an insufficient history or a user with a history of only a specific category. To address these problems, we propose a federated tree based on categories to predict generalized preferences. As the federation tree is generated based on the historical data of all users, it includes preferences for all categories. The proposed system facilitates the inference of the generalized preference score via the federated tree. When a federated tree is created using all historical data, the size of the tree increases, thereby incurring a long runtime. In this study, the historical data were sampled individually with a probability of ρ for all user histories to create a federated tree. Figure 5 depicts the average runtime and error of the preference score between the federated tree generated using all the data and that of the federated tree sampled according to ρ. We verified that the error in preference prediction is not significant despite generating a federated tree based on only sampled data rather than all historical data.

### 3.5. Similarity Tree

For similarity-based CF between user preference trees, we developed the similarity relationship as a graph. A CSG used to define similarity groups is depicted in Figure 6. The vertices of the graph correspond to the preference tree for each user, and the weight of the edge connecting two vertices indicates the similarity between the preference trees. The complete graph facilitates the prediction of the similarity for all users. However, it is computation-intensive because the spatial complexity for n vertices is $O\left(n^{2}\right)$.

In this study, we propose a CSG based on the maximum-spanning tree (MST) to construct a graph of similarity between users, as illustrated in Figure 6. The similarity S between the preference trees is calculated as the sum of the products of the preference scores of all commonly existing nodes in the preference trees of two users. Therefore, when the numbers of nodes in the preference tree of users A and B are N and M, respectively, the similarity $S_{A,B}$ is calculated with the time complexity $O\left(N\times \log_{2}M\times \log_{2}M\right)$. Algorithm 1 outlines the similarity score procedure. In a similar graph of the MST structure, it is possible to rapidly search for groups to perform CF based on the similarity threshold γ.
**Algorithm 1.** Computing similarity between users A and B**Input:**$\mathrm{Preference}\;\mathrm{tree}\;Tree^{A}$$\;\mathrm{and}\;Tree^{B}$**Output:**$S_{A,B}$1:**Initialize**$S_{A,B}$ to 02:**procedure**$\;\mathrm{getSimilar}(Tree^{A}$) **do**3:  let S be a stack4:  **for all**
$\;\mathrm{node}\;\mathrm{in}\;Tree^{A}$
5:    S.push(node)6:  **while** S is not empty **do**7:    $n_{c}^{A}=$ S.pop()8:    **find**
$\;n_{c}^{B}$
**in**
$\;Tree^{B}$
9:    **if**
$n_{c}^{B}$ is **exist**10:      
$S_{A,B}\;\leftarrow \;S_{A,B}+n_{c}^{A}\times n_{c}^{B}$
11:**return**$S_{A,B}$

### 3.6. Recommendation Algorithm

Existing recommendation systems operate based on the top-*N* method to determine the contents to be recommended when a user is offline. Consequently, it is difficult for the top-*N* method to satisfy real-time requirements. To realize real-time recommendation, our proposed recommendation algorithm is based on the personalized tree, federated tree, and CSG. The user-preference score for content, P, is computed using Equation (2):(2)$$P=\beta \times \left(F_{lcc}+CS_{lcc}\right)+\left(1-\beta \right)\times P_{lcc}$$
where $F_{lcc}$ and $CS_{lcc}$ denote the predicted preference scores for the LCC node in the federated tree and CSG, respectively. As $F_{lcc}$ and $P_{lcc}$ indicate the preferences for the *LCC* nodes in each tree, they are calculated using Equation (1). $CS_{lcc}$ is the average of $P_{lcc}$ in the similarity trees of a collaborative group exhibiting a similarity of γ or higher to the target user, as indicated in Equation (3):(3)$$CS_{lcc}=\frac{1}{N}\times \overset{N}{\underset{i=1}{\sum }}P_{lcc}^{i}$$
where N denotes the number of users in the similarity group and $P_{lcc}^{i}$ is the $P_{lcc}$ of the ith user. To recommend content to the user in real time, the recommendation is determined on the basis of an output sigmoid activation function for input P. The sigmoid output $O_{sig}$ for P is calculated using Equation (4). In this case, the content is recommended to the user only if $O_{sig}$ is greater than 0.5.
(4)$$O_{sig}=\frac{1}{1+e^{-P}}$$

## 4. Experimental Evaluation

### 4.1. Dataset

Experiments were performed using large-scale real-world Amazon product data [41,42]. The Amazon dataset comprised 24 primary categories, each including item attributes, historical data, and review data. In this study, we used the “electronic” category, which comprised 2673 users, 130,054 reviews, and 55,101 items as the dataset. The dataset was classified into 1541 categories, and the rating scores assigned to the items varied between one and five points. We defined items with rating scores between one and two points as low-preference items and those with scores between three and five points as high-preference items. To evaluate the performance of the proposed system, we used approximately 100,000 data points to generate the preference-tree model, and the remaining data were used for testing.

### 4.2. Evaluation Metrics

We evaluated four metrics—precision, recall, *F*1-*measure*, and accuracy—defined by Equations (5)–(8), respectively [43]. A true positive (*TP*) refers to the case in which high-preference content was recommended, and a true negative (*TN*) refers to the case in which low-preference content was not recommended. Conversely, a false positive (*FP*) indicates a recommendation of low-preference content, and a false negative (*FN*) indicates content with a high preference not being recommended. The precision denotes the proportion of predicted positive cases that are correctly identified as *TP*s. The recall is the proportion of TP cases that is correctly predicted, indicating the hit ratio of the recommendation results. The *F*1-*measure* is the harmonic mean of both the precision and recall and, in this study, represents a performance evaluation considering the trade-off between precision and recall.
(5)$$\mathrm{Precision}=\frac{TP}{TP+FP}$$
(6)$$\mathrm{Recall}=\frac{TP}{TP+FN}$$
(7)$$F1-measure=\frac{2\times Precision\times Recall}{Precision+Recall}$$
(8)$$\mathrm{Accuracy}=\frac{TP+TN}{TP+TN+FP+FN}$$

Furthermore, we evaluated novelty, which is the proportion of unknown items that do not exist in a personalized tree over the total number of items in the recommendation list. A total of N items were recommended; if the personalized preference for the ith item is $P_{lcc}^{i}$, novelty can be calculated using Equation (9):(9)$$\mathrm{Novelty}=\frac{\sum_{i=1}^{N}P_{lcc}^{i}}{N}$$

### 4.3. Experimental Environment

To evaluate the performance of the proposed recommendation system, we conducted a comparative experiment with max-heap-tree-, collaborative-, knowledge-, and MF-based recommendation systems. The experiments were performed on a desktop computer equipped with an Intel Core i7-10700K CPU and 16 GB of RAM. The MF-based recommendation system was implemented using the PyTorch framework, whereas the other schemes were implemented using the Eclipse framework. To conduct comparative experiments using the knowledge CF recommendation system, we designed an ontology based on a decision tree using Protégé [44]. Furthermore, we used the Jena framework to develop semantic functions [45].

### 4.4. Results

We conducted the first experiment by changing the constant α to determine whether the proposed preference tree is suitable for the recommendation system. Table 2 lists the average of the metrics measured based on the value of α. In this experiment, the values of β and γ were set to 0.5. The two factors affecting the similarity prediction are breadth and depth, which indicate diversity and granularity, respectively. As the value of α increased and decreased, the recommendation system focused on diversity and granularity, respectively. We observed that these two factors with α = 0.5 exhibited the best performance when the effects on the preference prediction were balanced. This observation validated that the proposed system performs best when the diversity and granularity of trees are considered at the same rate during the prediction of preferences. Therefore, the proposed preference-tree model is not biased toward diversity or granularity. This result confirms that the proposed preference-tree model is suitable for predicting user preferences.

In the second experiment, we evaluated the performance of the similarity tree, which prevents the recommendation system from overfitting the historical data. Table 3 lists the average metrics based on the similarity threshold γ. During the experiments, the values of α and β were set to 0.5. When γ = 0.4, the similarity tree included a preference tree with a low relevance. As the value of γ was small, a personalized tree with a low similarity was included in the preference prediction. Therefore, a wide range of recommendations could be made. We observed that the novelty was highest when γ was 0.4. Conversely, when γ was 0.7, precision and recall exhibited their best performances because the recommendation system considered a similarity tree. However, as the personalized tree significantly influences the preference prediction, the novelty performance was lowest when γ was set to a large value. In subsequent experiments, γ was set to 0.5 to ensure that the recommendation system could balance the user preference characteristics and novelty.

In the third experiment, we evaluated the effect of the proportional constant β of the personalized, federated, and similarity trees on the performance of the recommendation system. Federated and similarity trees predict generalized preferences, whereas personalized trees predict personalized preferences. Table 4 summarizes the average measured values of the five metrics based on the value of β. As was confirmed in the second experiment, the federated and similarity trees mitigated the overfitting problem by reducing the influence of personalized recommendations. Additionally, as the value of β decreased, the performance of the novelty was low while that of the recall was high. Based on previous experiments, we confirmed that the recommendation system ensures novelty while maintaining a high accuracy at α = γ = 0.5 and β = 0.6.

To evaluate the performance of the proposed recommendation system, we conducted comparative experiments using MF-, max-heap-tree-, and knowledge-based recommendation systems. Furthermore, we conducted an ablation study to evaluate the performance of the proposed federated, personalized preference, and similarity trees. The MF- and knowledge-based schemes measure metrics using the top-*N* method, with which *N* is set to 20.

Table 5 summarizes the performance evaluated based on the five metrics of the compared recommendation schemes. Proposed Scheme A exhibited a low novelty performance because it predicts preferences based on a personalized preference tree. As Proposed Schemes B and C predict preferences based on federated and similarity trees, respectively, the novelty and recall performance improved in comparison with that of Proposed Scheme A. These results indicate that an overfitting problem may occur when the recommendation system considers only the personalized preference tree. As the max-heap-tree-based system predicts preferences based on the maximum preference score for each node, it has difficulty considering various factors during the preference prediction. Consequently, the max-heap-tree-based scheme exhibited the lowest performance for all the metrics. The MF-based recommendation system has better novelty and accuracy performances than Proposed Scheme A because it predicts the preference based on only generalized preferences. However, both the max-heap-tree- and MF-based methods are unsuitable for real-time recommendations because they recommend items using the top-N method. The knowledge-based scheme predicts preferences based on the TF-IDF similarity vector between items; it exhibits the best performance in terms of precision. However, as the prediction of preferences is impossible for items that do not exist in the TF-IDF vector, we did not measure its novelty performance. In the case of Proposed Scheme D, which predicts the preference based on all trees, its accuracy and novelty performances were the highest among the compared schemes. Based on these experimental results, we confirmed that recommendation systems using the three types of trees can recommend with high accuracies.

Finally, we evaluated the runtimes when updating the proposed tree models and predicting their preferences. The runtimes were measured using approximately 100,000 historical data points used in the previous experiment. Figure 7 illustrates the runtime results for Proposed Schemes A, B, C, and D. We observed that Proposed Scheme B used additional federated trees, which increased the runtime slightly compared with that of Proposed Scheme A. Proposed Scheme C exhibited a significantly increased runtime compared with that of Proposed Scheme A because it has to identify similarity groups and search for LCC nodes in similarity trees. In the case of Proposed Scheme D, the tree updating runtime was measured to be 9.11 s, which is the largest runtime as all federated, personalized, and similarity trees should be updated. However, Proposed Scheme D can be applied in a real-time recommendation system because it is possible to predict user preferences within approximately 0.13 s. Consequently, the proposed recommendation system satisfies real-time requirements and provides a better prediction performance than existing systems without additional computations or requirements.

## 5. Conclusions

In this study, we proposed a preference-tree-based real-time recommendation system to avoid information overload. Most existing recommendation systems suffer from a large overhead incurred in learning regression models used for preference prediction or in selecting groups of similar users. To realize real-time recommendations, we developed various tree models that can predict user preferences with fast runtimes using federated, personalized preference, and similarity trees. The proposed system predicts preferences based on two balance constants and one similarity threshold to recommend items with a high accuracy while balancing both generalized and personalized preferences. To achieve a time-variant recommendation system, we employ an ordered HashMap data structure for fast tree searching and updating. The results of comparative experiments conducted to determine the constant values and thresholds confirmed that the proposed system facilitates the accurate recommendation of items to a target user. Additionally, we observed that the novelty performance of the proposed system is improved when generalized preferences are predicted based on both federated and similarity trees. In conclusion, we verified that the proposed recommendation system satisfies real-time requirements and mitigates both the cold-start and overfitting problems that are generally observed in existing systems. However, the proposed recommendation system predicts preferences through fixed parameters. That is, it performs recommendations using the same algorithm without differentiating between new and existing users. In future work, we plan to design a novel algorithm that analyzes the patterns of historical user data and develop a highly accurate recommendation algorithm using dynamic parameters for federated and similarity trees.

## Figures and Tables

**Figure 1 entropy-24-00503-f001:**
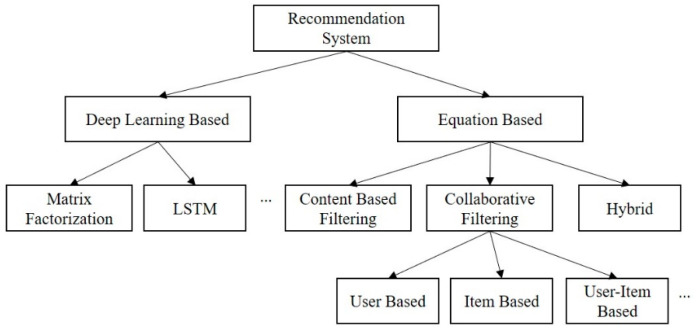
Classification of recommendation systems.

**Figure 2 entropy-24-00503-f002:**
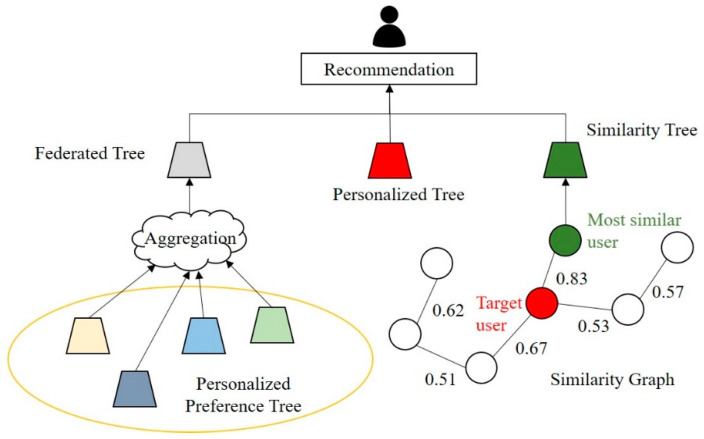
Proposed recommendation system architecture.

**Figure 3 entropy-24-00503-f003:**
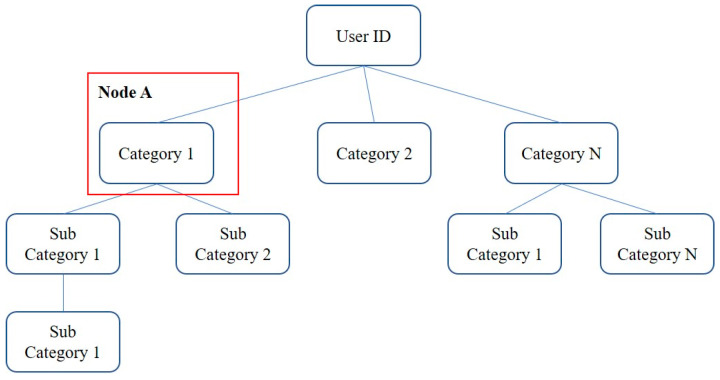
Structure of the proposed preference-tree model.

**Figure 4 entropy-24-00503-f004:**
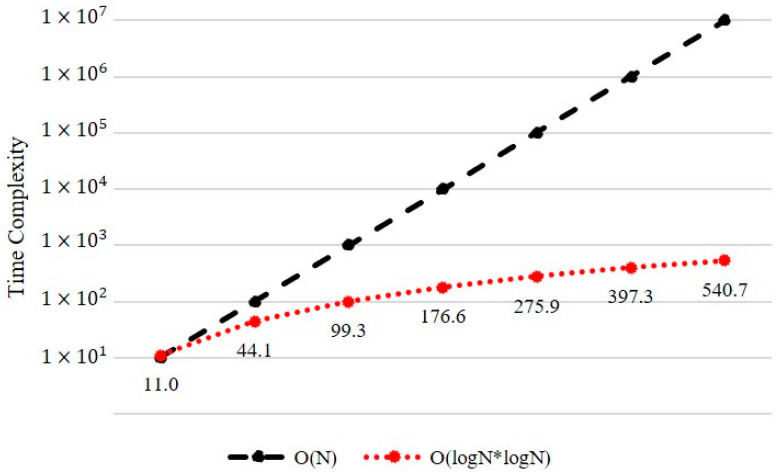
Time complexities of the data search with matrix and ordered HashMap.

**Figure 5 entropy-24-00503-f005:**
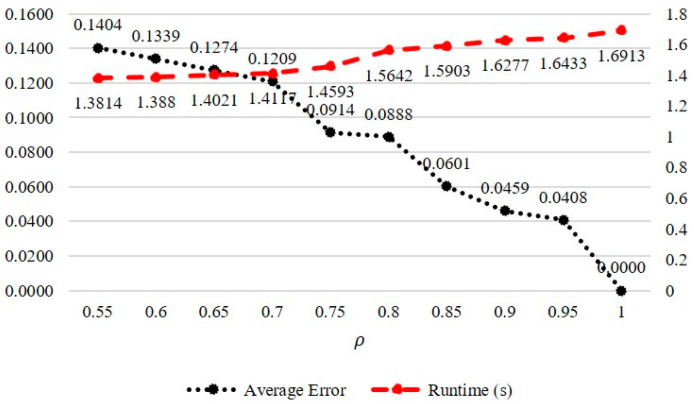
Average runtime and error of preference scores according to probability *ρ*.

**Figure 6 entropy-24-00503-f006:**
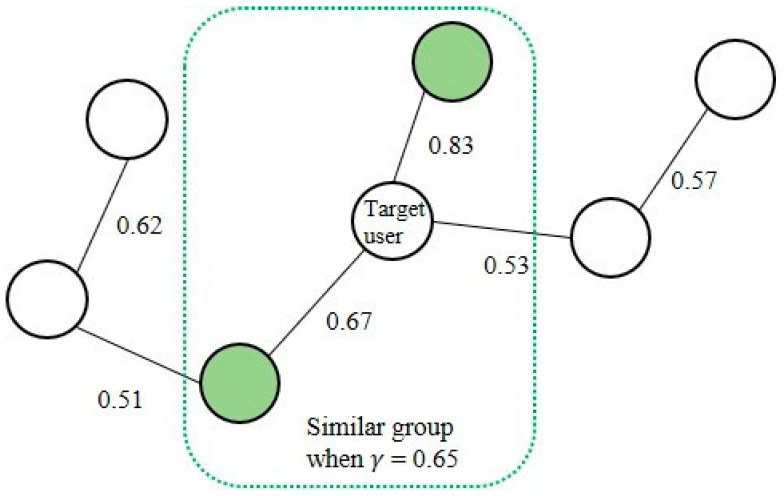
Example of collaborative similarity graph.

**Figure 7 entropy-24-00503-f007:**
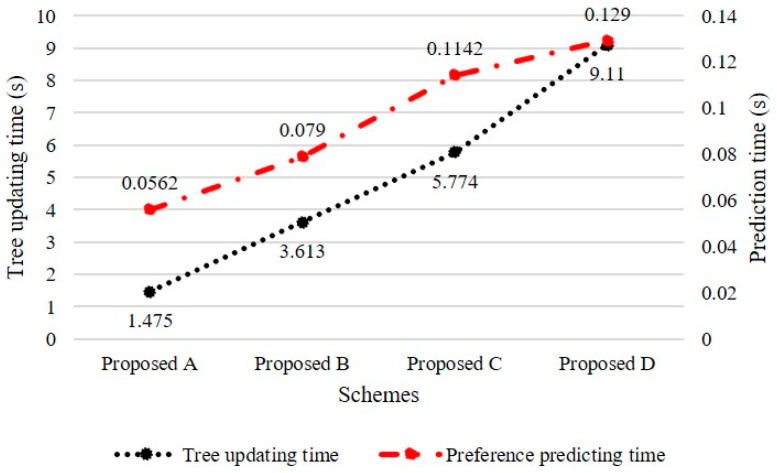
Execution time required for updating the tree and predicting preferences.

**Table 1 entropy-24-00503-t001:** Components of the preference-tree node.

Component	Description
c	Mapped category code of the node
$n_{c}$	Number of category *c*s in user’s historical data
$N_{child}$	Child nodes of the category *c* node
$d_{c}$	Depth of the category *c* node
$n_{d}$	Total number of categories at depth *d*
$d_{max}$	Maximum depth of the descendant node
$P_{c}$	User preference for the category *c* node
$w_{i}$	Weight of historical data in the *i*th section

**Table 2 entropy-24-00503-t002:** Average values of precision, recall, *F*1-*measure*, accuracy, and novelty measured based on α. The values of β and γ were set to 0.5.

α	Precision	Recall	*F*1-*Measure*	Accuracy	Novelty
0.3	0.659	0.713	0.685	0.596	0.285
0.4	0.633	0.737	0.681	0.579	0.224
0.5	0.646	0.725	0.683	0.588	0.302
0.6	0.635	0.739	0.683	0.570	0.331
0.7	0.620	0.739	0.674	0.571	0.275
0.8	0.604	0.689	0.644	0.543	0.292

**Table 3 entropy-24-00503-t003:** Average values of precision, recall, *F*1-*measure*, accuracy, and novelty measured based on γ. The values of α and β were set to 0.5.

γ	Precision	Recall	*F*1-*Measure*	Accuracy	Novelty
0.4	0.584	0.661	0.620	0.545	0.329
0.5	0.601	0.685	0.641	0.578	0.302
0.6	0.612	0.747	0.673	0.582	0.285
0.7	0.619	0.751	0.678	0.589	0.224

**Table 4 entropy-24-00503-t004:** Average values of precision, recall, *F*1-*measure*, accuracy, and novelty measured based on β. The values of α and γ were set to 0.5.

β	Precision	Recall	*F*1-*Measure*	Accuracy	Novelty
0.3	0.572	0.739	0.645	0.550	0.234
0.4	0.598	0.742	0.662	0.556	0.299
0.5	0.601	0.685	0.641	0.578	0.372
0.6	0.606	0.709	0.653	0.588	0.385
0.7	0.601	0.699	0.647	0.572	0.424
0.8	0.625	0.656	0.640	0.559	0.422

**Table 5 entropy-24-00503-t005:** Average values of precision, recall, *F*1-*measure*, accuracy, and novelty measured with α = γ = 0.5 and β = 0.6. Proposed Scheme A uses personalized tree. Proposed Scheme B uses personalized and federated trees. Proposed Scheme C uses personalized and similarity trees. Proposed Scheme D uses personalized, federated, and similarity trees.

Scheme	Precision	Recall	*F*1-*Measure*	Accuracy	Novelty
Proposed Scheme A	0.581	0.704	0.637	0.513	0.188
Proposed Scheme B	0.554	0.759	0.640	0.588	0.341
Proposed Scheme C	0.583	0.722	0.645	0.529	0.332
Proposed Scheme D	0.606	0.709	0.653	0.588	0.385
MF-based	0.590	0.682	0.633	0.544	0.229
Max-heap-tree-based	0.553	0.611	0.581	0.467	0.113
Knowledge-based	0.718	0.633	0.673	0.575	NAN

## Data Availability

Not applicable.

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
