# Peer review of "Preference-Tree-Based Real-Time Recommendation System"

_entropy, 2022, doi:10.3390/e24040503_

Round 1
Reviewer 1 Report
I have read the revised paper. The authors have addressed my comments and the quality of the paper has been improved. The paper can be accepted to publish in Entropy.Reviewer 2 Report
The manuscript has been significantly improved.
Reviewer 3 Report
The article is well written.
The comments, highlighted in my previous review, have been addressed in this version.
The article can be accepted in its current form.
This manuscript is a resubmission of an earlier submission. The following is a list of the peer review reports and author responses from that submission.
Round 1
Reviewer 1 Report
The article under consideration presents a “Real-Time Recommendation System” which uses various tree models that can predict user preferences with a fast runtime.
The article is very well-written and easy to follow. The quality is high from presentation point of view. The contributions are satisfactory and validation procedure is sound.
The detailed analysis for each section is given below:
(1) Abstract:
The problem and its importance have been described. The limitations of existing practices are clearly mentioned. The preposed solution is well-connected with the identified research gap. The only issue here is that how the authors have validated their solution (case/study or benchmarking) ?
(2) Introduction:
This section presents the research problem by explaining what is really required in the selected research area. It briefly overviews the existing practices. Limitations of existing practices have been identified and hence highlighting the research gape. According to the authors, the novelty of the proposed system is that it considers real-time user interactions. In other words, the system is updated in real-time to predict user preferences which makes it scalable to incorporate various factors. However, this claim is required to be rechecked and must be elaborated in detail so that the contributions towards the body of knowledge can be judged. Furthermore, it is required to mention that how authors have validated the solution (description of case studies and/or benchmarks. Why these benchmarks/case studies are important and interesting. What is the motivation behind the selection of these benchmarks). Finally, it is better to summarize the achieved outcomes and significance at the end of Introduction section.
(3) Related Work:
This section describes the existing methods and classify them in some categories. It is suggested to provide a brief discussion (at the end of this section) that how the proposed system is different from the categories, defined in this section.
(4) Proposed System:
The system structure and behavior are very well presented. The only point here is to illustrate or emphasize the fact that how the proposed system is “real-time” or “scalable” or “dynamic” ?? . The following article is an example of adaptive real-time systems: “Determination of Worst-Case Data using an Adaptive Surrogate Model for Real-Time System, Journal of Circuits, Systems and Computers, vol. 29, no.1, January 2020”. Similarly, it is also required to mention weather authors have evaluated various prediction models for performance comparison. The evaluation of various prediction models is given in the following article: “Estimating WCET using prediction models to compute fitness function of a genetic algorithm, Real-Time Systems, vol. 56, no. 3, pp. 28-63, July 2020.”. The authors are suggested to mention that how their proposed dynamic real-time prediction system is different from aforementioned articles?.
(5) Experiments and Evaluation : This section is very nicely presented. The experimental setup, description of tools and particular settings used in the experiment , description of benchmarks and case study, experimentation procedure and results have been provided. It is better to provide a discussion section that can comment on the obtained results (compared with state-of-art solutions). It is VERY IMPORTANT to highlight the strengths of your article as compared to existing methods. Discuss the reasons that why you were able to obtain good results. Similarly, it is equally important to mention the shortcomings of your solution with respect to current methods. Discuss the reasons for these limitations and shortcomings.
Reviewer 2 Report
The paper is not well written and there are multiple mistakes in several sections. The use of English should be also improved. Many sentences are not clear and confuse the reader, for example starting with the abstract, the sentence “However, information filtering-based recommendation systems may cause data sparsity and cold start problems.” is confusing. Challenges such as data sparsity or the cold-start problem are inherently connected to the data and not generated/caused by a recommender system. The authors should rephrase or elaborate on their view.
The two major weaknesses of this papers are the lacking novelty as well as the lack of a thorough evaluation study. Although the authors have made an effort to tackle long-standing and difficult problems in the field of recommender systems, their contribution is not substantial. Additionally, the authors have not compared their approach against any recommender system, neither baseline nor state of the art one. In the literature, there is a plethora of different approaches and plenty of them are accompanied with publicly available code, facilitating this way their use by other scientists. Last, the authors used only a single dataset to validate their approach.
Reviewer 3 Report
In this paper, the authors proposed a preference tree-based real-time recommendation system. They used various tree models to predict user preferences with a fast runtime hence the system can provide real-time recommendations. The proposed system predicts preferences based on two balance constants and one similarity threshold to recommend contents with high accuracy while balancing generalized and personalized preferences. The experiments were conducted on the large-scale real-world Amazon product dataset. The proposed model also compared with some existing models and the proposed model outperforms the other models.
This is an interesting paper, and the overall work is good. Here are some minor comments for your consideration:
- The authors should state what are the scientific contributions of this research project in Section 1 Introduction.
- It would be appreciated if authors could discuss any limitations of the proposed model and how to improve the model in the future.
